# Robust covariance estimation with missing values and cell-wise contamination

**Karim Lounici**
CMAP
Ecole Polytechnique
Palaiseau, France
karim.lounici@polytechnique.edu

**Gregoire Pacreau**
CMAP
Ecole Polytechnique
Palaiseau, France
gregoire.pacreau@polytechnique.edu

## Abstract

Large datasets are often affected by cell-wise outliers in the form of missing or erroneous data. However, discarding any samples containing outliers may result in a dataset that is too small to accurately estimate the covariance matrix. Moreover, the robust procedures designed to address this problem require the invertibility of the covariance operator and thus are not effective on high-dimensional data. In this paper, we propose an unbiased estimator for the covariance in the presence of missing values that does not require any imputation step and still achieves near minimax statistical accuracy with the operator norm. We also advocate for its use in combination with cell-wise outlier detection methods to tackle cell-wise contamination in a high-dimensional and low-rank setting, where state-of-the-art methods may suffer from numerical instability and long computation times. To complement our theoretical findings, we conducted an experimental study which demonstrates the superiority of our approach over the state of the art both in low and high dimension settings.

## 1 Introduction

Outliers are a common occurrence in datasets, and they can significantly affect the accuracy of data analysis. While research on outlier detection and treatment has been ongoing since the 1960s, much of it has focused on cases where entire samples are outliers (Huber's contamination model) [11, 41, 13]. While sample-wise contamination is a common issue in many datasets, modern data analysis often involves combining data from multiple sources. For example, data may be collected from an array of sensors, each with an independent probability of failure, or financial data may come from multiple companies, where reporting errors from one source do not necessarily impact the validity of the information from the other sources. Discarding an entire sample as an outlier when only a few features are contaminated can result in the loss of valuable information, especially in high-dimensional datasets where samples are already scarce. It is important to identify and address the specific contaminated features, rather than simply treating the entire sample as an outlier. In fact, if each dimension of a sample has a contamination probability of $\varepsilon$, then the probability of that sample containing at least one outlier is given by $1 - (1 - \varepsilon)^p$, where $p$ is the dimensionality of the sample. In high dimension, this probability can quickly exceed $50\%$, surpassing the breakdown point of many robust estimators designed for the Huber sample-wise contamination setting. Hence, it is crucial to develop robust methods that can handle cell-wise contaminations and still provide accurate results.

The issue of cell-wise contamination, where individual cells in a dataset may be contaminated, was first introduced in [3]. However, the issue of missing data due to outliers was studied much earlier, dating back to the work of [35]. Although missing values in a dataset are much easier to detect than outliers, they can lead to errors in estimating the location and scale of the underlying distribution [25]

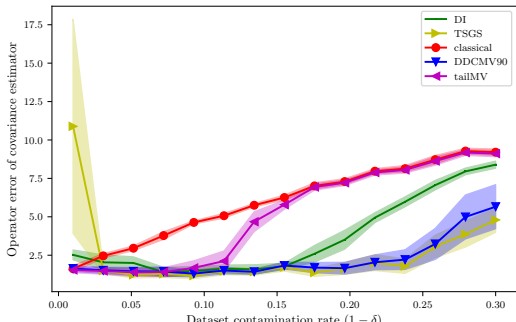

| Estimator | Computation time | | Matrix inversion |
|---|---|---|---|
| | p=50 | p=100 | |
| tailMV | $10^{-3} \pm 10^{-4}$ | $10^{-2} \pm 10^{-3}$ | no |
| DDCMV | $0.6 \pm 10^{-3}$ | $0.7 \pm 0.007$ | no |
| DI | $6 \pm 0.5$ | $74 \pm 4$ | yes |
| TSGS | $20 \pm 0.2$ | $200 \pm 10$ | yes |

Figure 1: Left: Estimation error of the covariance matrix for $n = 100$, $p = 50$, $\mathbf{r}(\Sigma) = 2$ under a Dirac contamination (`tailMV` and `DDCMV` are our methods). Here $\varepsilon = 1$ and $\delta$ varies in $(0, 1)$. Right: For each method, mean computation time (in seconds) over 20 repetitions and whether it uses matrix inversion. For $p = 100$, we had to raise $r(\Sigma)$ to 10 otherwise both `DI` and `TSGS` would fail due to numerical instability.

and can negatively affect the performance of supervised learning algorithms [17]. This motivated the development of the field of data imputation. Several robust estimation methods have been proposed to handle missing data, including Expectation Maximization (EM)-based algorithms [7], maximum likelihood estimation [14] and Multiple Imputation [25], among which we can find k-nearest neighbor imputation [39] and iterative imputation [42]. Recently, sophisticated solutions based on deep learning, GANs [47, 28, 9], VAE [27] or Diffusion schemes [48] have been proposed to perform complex tasks like artificial data generation or image inpainting. The aforementioned references focus solely on minimising the entrywise error for imputed entries. Noticeably, our practical findings reveal that applying state-of-the-art imputation methods to complete the dataset, followed by covariance estimation on the completed dataset, does not yield satisfactory results when evaluating the covariance estimation error using the operator norm.

In comparison to data missingness or its sample-wise counterpart, the cell-wise contamination problem is less studied. The Detection Imputation (DI) algorithm of [32] is an EM type procedure combining a robust covariance estimation method with an outlier detection method to iteratively update the covariance estimation. Other methods include adapting methodology created for Huber contamination for the cell-wise problem, such as in [6] or [2]. In high dimensional statistics, however, most of these methods fail due to high computation time and numerical instability. Or they are simply not designed to work in this regime since they are based on the Mahalanobis distance, which requires an inversion of the estimated covariance matrix. This is a major issue since classical covariance matrix estimators have many eigenvalues close to zero or even exactly equal to zero in high-dimension. To the best of our knowledge, no theoretical result exists concerning the statistical accuracy of these methods in the cell-wise contamination setting contrarily to the extensive literature on Huber's contamination [1].

**Contributions.** In this paper we address the problem of high-dimensional covariance estimation in the presence of missing observations and cell-wise contamination. To formalize this problem, we adopt and generalize the setting introduced in [10]. We propose and investigate two different strategies, the first based on filtering outliers and debiasing and the second based on filtering outliers followed by imputation and standard covariance estimation. We propose novel computationally efficient and numerically stable procedures that avoid matrix inversion, making them well-suited for high-dimensional data. We derive non-asymptotic estimation bounds of the covariance with the operator norm and minimax lower bounds, which clarify the impact of the missing value rate and outlier contamination rate. Our theoretical results also improve over [26] in the MCAR and no contamination. Next, we conduct an experimental study on synthetic data, comparing our proposed methods to the state-of-the-art (SOTA) methods. Our results demonstrate that SOTA methods fail in the high-dimensional regime due to matrix inversions, while our proposed methods perform well in this regime, highlighting their effectiveness. Then we demonstrate the practical utility of our approach by applying it to real-life datasets, which highlights that the use of existing estimation methods significantly alters the spectral properties of the estimated covariance matrices. This implies

that cell-wise contamination can significantly impact the results of dimension reduction techniques like PCA by completely altering the computed principal directions. Our experiments demonstrate that our methods are more robust to cell-wise contamination than SOTA methods and produce reliable estimates of the covariance.

## 2  Missing values and cell-wise contamination setting

Let $X_1, \ldots, X_n$ be $n$ i.i.d. copies of a zero mean random vector $X$ admitting unknown covariance operator $\Sigma = \mathbb{E}\left[X \otimes X\right]$, where $\otimes$ is the outer product. Denote by $X_i^{(j)}$ the $j$th component of vector $X_i$ for any $j \in [p]$. All our results are non-asymptotic and cover a wide range of configurations for $n$ and $p$ including the high-dimensional setting $p \gg n$. In this paper, we consider the following two realistic scenarios where the measurements are potentially corrupted.

**Missing values.**  We assume that each component $X_i^{(j)}$ is observed independently from the others with probability $\delta \in (0, 1]$. Formally, we observe the random vector $Y \in \mathbb{R}^p$ defined as follows:

$$Y_i^{(j)} = d_{i,j} X_i^{(j)}, 1 \le i \le n, 1 \le j \le p \tag{1}$$

where $d_{ij}$ are independent realisations of a bernoulli random variable of parameter $\delta$. This corresponds the Missing Completely at Random (MCAR) setting of [35]. Our theory also covers the more general Missing at Random (MAR) setting in Theorem 2.

**Cell-wise contamination.**  Here we assume that some missing components $X_i^{(j)}$ can be replaced with probability $\varepsilon$ by some independent noise variables, representing either a poisoning of the data or random mistakes in measurements. The observation vector $Y$ then satisfies:

$$Y_i^{(j)} = d_{i,j} X_i^{(j)} + (1 - d_{i,j}) e_{i,j} \xi_i^{(j)}, 1 \le i \le n, 1 \le j \le p \tag{2}$$

where $\xi_i^{(j)}$ are independent erroneous measurements and $e_{i,j}$ are i.i.d. bernoulli random variables with parameter $\varepsilon$. We also assume that all the variables $X_i, \xi_i^{(j)}, d_{i,j}, e_{i,j}$ are mutually independent. In this scenario, a component $X_i^{(j)}$ is either perfectly observed with probability $\delta$, replaced by a random noise with probability $\varepsilon' = \varepsilon(1 - \delta)$ or missing with probability $(1 - \delta)(1 - \varepsilon)$. Cell-wise contamination as introduced in [3] corresponds to the case where $\varepsilon = 1$, and thus $\varepsilon' = 1 - \delta$.

In both of these settings, the task of estimating the mean of the random vectors $X_i$ is well-understood, as it reduces to the classical Huber setting for component-wise mean estimation. One could for instance apply the Tuker median on each component separately [3]. However, the problem becomes more complex when we consider non-linear functions of the data, such as the covariance operator. Robust covariance estimators originally designed for the Huber setting may not be suitable when applied in the presence of missing values or cell-wise contaminations.

We study a simple estimator based on a correction of the classical covariance estimator on $Y_1, \ldots, Y_n$ as introduced in [26] for the missing values scenario. The procedure is based on the following observation, linking $\Sigma^Y$ the covariance of the data with missing values and $\Sigma$ the true covariance:

$$\Sigma = \left(\delta^{-1} - \delta^{-2}\right) \operatorname{diag}(\Sigma^Y) + \delta^{-2} \Sigma^Y \tag{3}$$

Note that this formula assumes the knowledge of $\delta$. In the missing values scenario, $\delta$ can be efficiently estimated by a simple count of the values exactly set to 0 or equal to `NaN` (not a number). In the contamination setting (2), the operator $\Sigma^Y = \mathbb{E}\left(Y \otimes Y\right)$ satisfies, for $\Lambda = \mathbb{E}\left[\xi \otimes \xi\right]$:

$$\Sigma^Y = \delta^2 \Sigma + (\delta - \delta^2) \operatorname{diag}(\Sigma) + \varepsilon(1 - \delta)\Lambda.$$

In this setting, as one does not know the exact location and number of outliers we propose to estimate $\delta$ by the proportion of data remaining after the application of a filtering procedure.

**Notations.**  We denote by $\odot$ the Hadamard (or term by term) product of two matrices and by $\otimes$ the outer product of vectors, i.e. $\forall x, y \in \mathbb{R}^d, x \otimes y = xy^\top$. We denote by $\|.\|$ and $\|.\|_F$ the operator and Frobenius norms of a matrix respectively. We denote by $\|\cdot\|_2$ the vector $l_2$-norm.

# 3 Estimation of covariance matrices with missing values

We consider the scenario outlined in (1) where the matrix $\Sigma$ is of approximately low rank. To quantify this, we use the concept of effective rank, which provides a useful measure of the inherent complexity of a matrix. Specifically, the effective rank of $\Sigma$ is defined as follows

$$\boldsymbol{r}(\Sigma) := \frac{\mathbb{E}\left\|X\right\|_2^2}{\|\Sigma\|} = \frac{\operatorname{tr}(\Sigma)}{\|\Sigma\|} \tag{4}$$

We note that $0 \leq \boldsymbol{r}(\Sigma) \leq \operatorname{rank}(\Sigma)$. Furthermore, for approximately low rank matrices with rapidly decaying eigenvalues, we have $\boldsymbol{r}(\Sigma) \ll \operatorname{rank}(\Sigma)$. This section presents a novel analysis of the estimator defined in equation (3), which yields a non-asymptotic minimax optimal estimation bound in the operator norm. Our findings represent a substantial enhancement over the suboptimal guarantees reported in [26, 18]. Similar results could be established for the Frobenius norm using more straightforward arguments, as those in [4] or [31]. We give priority to the operator norm since it aligns naturally with learning tasks such as PCA. See [19, 21, 22] and the references cited therein.

We need the notion of Orlicz norms. For any $\alpha \geq 1$, the $\psi_\alpha$-norms of a real-valued random variable $V$ are defined as: $\|V\|_{\psi_\alpha} = \inf\{u > 0, \mathbb{E}\exp\left(|V|^\alpha/u^\alpha\right) \leq 2\}$. A random vector $X \in \mathbb{R}^p$ is sub-Gaussian if and only if $\forall x \in \mathbb{R}^p, \|\langle X, x\rangle\|_{\psi_2} \lesssim \|\langle X, x\rangle\|_{L^2}$.

**Minimax lower-bound.** We now provide a minimax lower bound for the covariance estimation with missing values problem. Let $\mathcal{S}_p$ the set of $p \times p$ symmetric semi-positive matrices. Then, define $\mathcal{C}_{\overline{r}} = \{S \in \mathcal{S}_p : \boldsymbol{r}(S) \leq \overline{r}\}$ the set of matrices of $\mathcal{S}_p$ with effective rank at most $\overline{r}$.

**Theorem 1.** *Let $p, n, \overline{r}$ be strictly positive integers such that $p \geq \max\{n, 2\overline{r}\}$. Let $X_1, \ldots, X_n$ be i.i.d. random vectors in $\mathbb{R}^p$ with covariance matrix $\Sigma \in \mathcal{C}_{\overline{r}}$. Let $(d_{i,j})_{1 \leq i \leq n, 1 \leq j \leq p}$ be an i.i.d. sequence of Bernoulli random variables with probability of success $\delta \in (0, 1]$, independent from the $X_1, \ldots, X_n$. We observe $n$ i.i.d. vectors $Y_1, \ldots, Y_n \in \mathbb{R}^p$ such that $Y_i^{(j)} = d_{i,j} X_i^{(j)}$, $i \in [n]$, $j \in [p]$. Then there exists two absolute constants $C > 0$ and $\beta \in (0, 1)$ such that:*

$$\inf_{\widehat{\Sigma}} \max_{\Sigma \in \mathcal{C}_{\overline{r}}} \mathbb{P}_\Sigma\left(\left\|\widehat{\Sigma} - \Sigma\right\| \geq C\frac{\|\Sigma\|}{\delta}\sqrt{\frac{\boldsymbol{r}(\Sigma)}{n}}\right) \geq \beta \tag{5}$$

*where $\inf_{\widehat{\Sigma}}$ represents the infimum over all estimators $\widehat{\Sigma}$ of matrix $\Sigma$ based on $Y_1, \ldots, Y_n$.*

*Sketch of proof.* We first build a sufficiently large test set of hard-to-learn covariance operators exploiting entropy properties of the Grassmann manifold such that the distance between any two distinct covariance operator is at least of the order $\frac{\|\Sigma\|}{\delta}\sqrt{\frac{\boldsymbol{r}(\Sigma)}{n}}$. Next, in order to control the Kullback-Leibler divergence of the observations with missing values, we exploit in particular interlacing properties of the eigenvalues of the perturbed covariance operators [37]. □

This lower bound result improves upon [26, Theorem 2] as it relaxes the hypotheses on $n$ and $\overline{r}$. More specifically, the lower bound in [26] requires $n \geq 2\overline{r}^2/\delta^2$ while we only need the mild assumption $p \geq \max\{n, 2\overline{r}\}$. Our proof leverages the properties of the Grassmann manifold, which has been previously utilized in different settings such as sparse PCA without missing values or contamination [45] and low-rank covariance estimation without missing values or contamination [23]. However, tackling missing values in the Grassmann approach adds a technical challenge to these proofs as they modify the distribution of observations. Our proof requires several additional nontrivial arguments to control the distribution divergences, which is a crucial step in deriving the minimax lower bound.

**Non-asymptotic upper-bound in the operator norm.** We provide an upper bound of the estimation error in operator norm. We write $Y_i = d_i \odot X_i$. Let $\widehat{\Sigma}^Y = n^{-1}\sum_{i=1}^n Y_i \otimes Y_i$ be the classical covariance estimator of the covariance of $Y$. When the dataset contains missing values and corruptions, $\widehat{\Sigma}^Y$ is a biased estimator of $\Sigma$. Exploiting Equation (3), [26] proposed the following unbiased estimator of the covariance matrix $\Sigma$:

$$\widehat{\Sigma} = \delta^{-2}\widehat{\Sigma}^Y + (\delta^{-1} - \delta^{-2})\operatorname{diag}\left(\widehat{\Sigma}^Y\right). \tag{6}$$

The following result is from [18, Theorem 4.2].

**Lemma 1.** *Let $X_1, \ldots, X_n$ be i.i.d. sub-Gaussian random variables in $\mathbb{R}^p$, with covariance matrix $\Sigma$, and let $d_{ij}, i \in [1, n], j \in [1, p]$ be i.i.d bernoulli random variables with probability of success $\delta > 0$. Then there exists an absolute constant $C$ such that, for $t > 0$, with probability at least $1 - e^{-t}$:*

$$\left\| \widehat{\Sigma} - \Sigma \right\| \leq C \left\| \Sigma \right\| \left( \sqrt{\frac{\boldsymbol{r}(\Sigma) \log \boldsymbol{r}(\Sigma)}{\delta^2 n}} \vee \sqrt{\frac{t}{\delta^2 n}} \vee \frac{\boldsymbol{r}(\Sigma)(t + \log \boldsymbol{r}(\Sigma))}{\delta^2 n} \log(n) \right) \tag{7}$$

This result uses a recent unbounded version of the non-commutative Bernstein inequality, thus yielding some improvement upon the previous best known bound of [26]. Theorem 1 and Lemma 1 provide some important insights on the minimax rate of estimation in the missing values setting. In the high-dimensional regime $p \geq \max\{n, 2\overline{r}\}$ and $n \geq \delta^{-2}\mathbf{r}(\Sigma)(\log \mathbf{r}(\Sigma)) \log^2 n$, we observe that the two bounds coincide up to a logarithmic factor in $\mathbf{r}(\Sigma)$, hence clarifying the impact of missing data on the estimation rate via the parameter $\delta$.

**Heterogeneous missingness.** We can extend the correction to the more general case where each feature has a different missing value rate known as the Missing at Random (MAR) setting in [35]. We denote by $\delta_j \in (0, 1]$ the probability to observe feature $X^{(j)}$, $1 \leq j \leq p$ and we set $\delta := (\delta_j)_{j \in [p]}$. As in the MCAR setting, the probabilities $(\delta_j)_{j \in [p]}$ can be readily estimated by tallying the number of missing entries for each feature. Hence they will be assumed to be known for the sake of brevity. Let $\delta_{\text{inv}} = (\delta_j^{-1})_{j \in [p]}$ be the vector containing the inverse of the observing probabilities and $\Delta_{\text{inv}} = \delta_{\text{inv}} \otimes \delta_{\text{inv}}$. In this case, the corrected estimator becomes :

$$\widehat{\Sigma} = \Delta_{\text{inv}} \odot \widehat{\Sigma}^Y + \left( \text{diag}\left( \delta_{\text{inv}} \right) - \Delta_{\text{inv}} \right) \odot \text{diag}\left( \widehat{\Sigma}^Y \right) \tag{8}$$

Let $\overline{\delta} = \max_j\{\delta_j\}$ and $\underline{\delta} = \min_j\{\delta_j\}$ be the largest and smallest probabilities to observe a feature.

**Theorem 2.** *(i) Let $X_1, \ldots, X_n$ be i.i.d. sub-Gaussian random variables in $\mathbb{R}^p$, with covariance matrix $\Sigma$. We consider the MAR setting described above. Then the estimator (8) satisfies, for any $t > 0$, with probability at least $1 - e^{-t}$*

$$\left\| \widehat{\Sigma} - \Sigma \right\| \leq C \left\| \Sigma \right\| \frac{\overline{\delta}}{\underline{\delta}^2} \left( \sqrt{\frac{\boldsymbol{r}(\Sigma) \log \boldsymbol{r}(\Sigma)}{n}} \vee \sqrt{\frac{t}{n}} \vee \frac{\boldsymbol{r}(\Sigma)(t + \log \boldsymbol{r}(\Sigma))}{\overline{\delta} n} \log n \right) \tag{9}$$

*(ii) Let $p, n, \overline{r}$ be strictly positive integers such that $p \geq \max\{n, 2\overline{r}\}$. Let $X_1, \ldots, X_n$ be i.i.d. random vectors in $\mathbb{R}^p$ with covariance matrix $\Sigma \in \mathcal{C}_{\overline{r}}$. Then,*

$$\inf_{\widehat{\Sigma}} \max_{\Sigma \in \mathcal{C}_{\overline{r}}} \mathbb{P}_{\Sigma} \left( \left\| \widehat{\Sigma} - \Sigma \right\| \geq C \frac{\left\| \Sigma \right\|}{\overline{\delta}} \sqrt{\frac{\boldsymbol{r}(\Sigma)}{n}} \right) \geq \beta. \tag{10}$$

If $\overline{\delta} \asymp \underline{\delta}$ then the rates for the MCAR and MAR settings match. The proof is a straightforward adaptation of the proof in the MCAR setting.

## 4 Optimal estimation of covariance matrices with cell-wise contamination

In this section, we consider the cell-wise contamination setting (2).We derive both an upper bound on the operator norm error of the estimator (6) and a minimax lower bound for this specific setting. Let us assume that the $\xi_1, \ldots \xi_n$ are sub-Gaussian r.v. Note also that $\Lambda := \mathbb{E}[\xi_1 \otimes \xi_1]$ is diagonal in the cell-wise contamination setting (2).

**Minimax lower-bound.** The lower bound for missing values still applies to the contaminated case as missing values are a particular case of cell-wise contamination. But we want a more general lower bound that also covers the case of adversarial contaminations.

**Theorem 3.** *Let $p, n, \overline{r}$ be strictly positive integers such that $p \geq \max\{n, 2\overline{r}\}$. Let $X_1, \ldots, X_n$ be i.i.d. random vectors in $\mathbb{R}^p$ with covariance matrix $\Sigma \in \mathcal{C}_{\overline{r}}$. Let $(d_{i,j})_{1 \leq i \leq n, 1 \leq j \leq p}$ be i.i.d. sequence of bernoulli random variables of probability of success $\delta \in (0, 1]$, independent to the $X_1, \ldots, X_n$. We observe $n$ i.i.d. vectors $Y_1, \ldots, Y_n \in \mathbb{R}^p$ satisfying (2) where $\xi_i$ are i.i.d. of arbitrary distribution $Q$. Then there exists two absolute constants $C > 0$ and $\beta \in (0, 1)$ such that:*

$$\inf_{\widehat{\Sigma}} \max_{\Sigma \in \mathcal{C}_{\overline{r}}} \max_{Q} \mathbb{P}_{\Sigma, Q} \left( \left\| \widehat{\Sigma} - \Sigma \right\| \geq C \frac{\left\| \Sigma \right\|}{\delta} \sqrt{\frac{\boldsymbol{r}(\Sigma)}{n}} \bigvee \frac{\varepsilon(1 - \delta)}{\delta} \right) \geq \beta \tag{11}$$

*where $\inf_{\widehat{\Sigma}}$ represents the infimum over all estimators of matrix $\Sigma$ and $\max_Q$ is the maximum over all contamination $Q$.*

The proof of this theorem adapts an argument developed to derive minimax lower bounds in the Huber contamination setting. See App. G.3 for the full proof.

**Non-asymptotic upper-bound in the operator norm.** Note that the term $\varepsilon(1-\delta)\Lambda$ in the cell-wise contamination setting is negligible when $\delta \approx 1$ or $\varepsilon \approx 0$. Using the DDC detection procedure of [32], we can detect the contaminations and make $\varepsilon$ smaller without decreasing $\delta$ too much. For simplicity, we assume from now on that the $\xi_i^{(j)}$ are i.i.d. with common variance $\sigma_\xi^2$. Hence $\Lambda = \sigma_\xi^2 I_p$. We further assume that the $\xi_i^{(j)}$ are sub-Gaussian since we observed in our experiments that filtering removed all the large-valued contaminated cells and only a few inconspicuous contaminated cells remained. Our procedure (6) satisfies the following result.

**Theorem 4.** *Let the assumptions of Theorem 1 be satisfied. We assume in addition that the observations $Y_1, \ldots, Y_n$ satisfy (2) with $\varepsilon \in [0,1)$ and $\delta \in (0,1]$ and i.i.d. sub-Gaussian $\xi_i^{(j)}$'s. Then, for any $t > 0$, with probability at least $1 - e^{-t}$:*

$$\left\| \widehat{\Sigma} - \Sigma \right\| \lesssim \|\Sigma\| \left( \sqrt{\frac{\boldsymbol{r}(\Sigma) \log \boldsymbol{r}(\Sigma)}{\delta^2 n}} \vee \sqrt{\frac{t}{\delta^2 n}} \vee \frac{\boldsymbol{r}(\Sigma)(t + \log \boldsymbol{r}(\Sigma))}{\delta^2 n} \log(n) \right) + \frac{\varepsilon(1-\delta)\sigma_\xi^2}{\delta}$$

$$+ \frac{(1-\delta)\varepsilon}{\delta^2 \sqrt{|\log((1-\delta)\varepsilon)|}} \sigma_\xi^2 \left( \sqrt{\frac{p}{n}} \vee \frac{p}{n} \vee \sqrt{\frac{t}{n}} \vee \frac{t}{n} \right)$$

$$+ D(\delta, p) \sqrt{\frac{t + \log(p)}{n}} + \sqrt{\delta(1-\delta)\varepsilon \sigma_\xi^2 p} \sqrt{\operatorname{tr}(\Sigma)} \log(n) \frac{t + \log(p)}{n},$$

*where $D(\delta, p) = \sqrt{\frac{(1-\delta)}{\delta^2}\varepsilon(p-2)\sigma_\xi^2 \left[ 2\|\Sigma\| + \sigma_\xi^2 \right] + \frac{(1-\delta)}{\delta^3}\varepsilon\sigma_\xi^4 \left( |\operatorname{tr}(\Sigma) - \delta(p-2)| + \|\Sigma\| \right)}$.*

See App F.3 for the proof. As emphasized in [20], the effective rank $\boldsymbol{r}(\Sigma)$ provides a measure of the statistical complexity of the covariance learning problem in the absence of any contamination. However, when cell-wise contamination is present, the statistical complexity of the problem may increase from $\boldsymbol{r}(\Sigma)$ to $\boldsymbol{r}(\Lambda) = p$. Fortunately, if the filtering process reduces the proportion of cell-wise contamination $\varepsilon$ such that $(1-\delta)\varepsilon \operatorname{tr}(\Lambda) \leq \delta \operatorname{tr}(\Sigma)$ and $\varepsilon \|\Lambda\| \leq \delta \|\Sigma\|$. Then we can effectively mitigate the impact of cell-wise contamination. Indeed, we deduce from Theorem 4 that

$$\left\| \widehat{\Sigma} - \Sigma \right\| \lesssim \|\Sigma\| \left( \sqrt{\frac{\boldsymbol{r}(\Sigma) \log \boldsymbol{r}(\Sigma)}{\delta^2 n}} \vee \sqrt{\frac{t}{\delta^2 n}} \vee \frac{\boldsymbol{r}(\Sigma)(t + \log \boldsymbol{r}(\Sigma))}{\delta^2 n} \log(n) \right) + \frac{\varepsilon(1-\delta)\sigma_\xi^2}{\delta}$$

$$+ \frac{1}{\delta}\sqrt{(1-\delta)\operatorname{tr}(\Sigma)} \sqrt{\frac{t + \log(p)}{n}} \left( \sqrt{\delta \, \sigma_\xi^2} + \sqrt{\operatorname{tr}(\Sigma)} \log(n) \sqrt{\frac{t + \log(p)}{n}} \right),$$
$$\tag{12}$$

where we considered for convenience the reasonable scenario where $\delta(p-2) \geq \operatorname{tr}(\Sigma)$ and $\sigma_\xi^2 \geq \|\Sigma\|$. The combination of the upper bound (12) with the lower bound in Theorem (3) provides the first insights into the impact of cell-wise contamination on covariance estimation.

## 5 Experiments

In our experiments, MV refers either to the debiased MCAR covariance estimator (6) or to its MAR extension (8). The synthetic data generation is described in App. A. We also performed experiments on real life datasets described in App. B. All experiments were conducted on a 2020 MacBook Air with a M1 processor (8 cores, 3.4 GHz). [1]

---

[1]Code available at `https://github.com/klounici/COVARIANCE_contaminated_data`

Table 1: Execution time of the covariance estimation procedures (in milliseconds) with $n = 300$ averaged over all values of the contamination rate $\delta$ and 20 repetitions.

| method | $p = 50$ | $p = 100$ | $p = 500$ |
|---|---|---|---|
| MV (ours) | $0.29 \pm 0.03$ | $0.49 \pm 0.08$ | $9.7 \pm 4.5$ |
| KNNImputer (KNN) | $26 \pm 9.8$ | $45 \pm 17$ | $470 \pm 190$ |
| IterativeImputer (II) | $940 \pm 350$ | $2,800 \pm 900$ | $3.7 \times 10^5 \pm 1.1 \times 10^5$ |
| Gain | $6,900 \pm 480$ | $1.1 \times 10^4 \pm 250$ | $8.8 \times 10^4 \pm 1.1 \times 10^3$ |
| MIWAE | $5.1 \times 10^4 \pm 2.8 \times 10^3$ | $6.7 \times 10^4 \pm 550$ | $1.77 \times 10^5 \pm 5.8 \times 10^3$ |

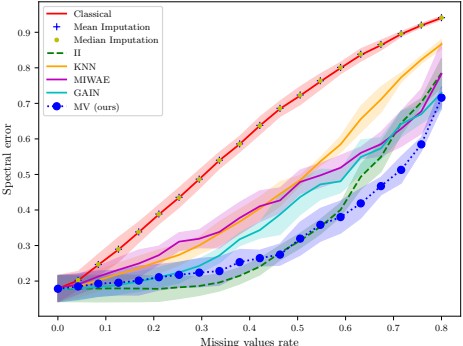

Figure 2: Estimation error on a synthetic dataset with $p = 50$, $n = 300$, $r(\Sigma) = 5$.

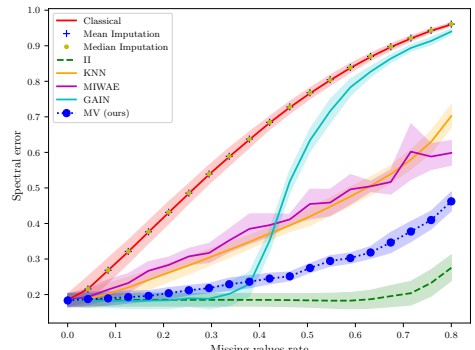

Figure 3: Estimation error on a synthetic dataset with $p = 500$, $n = 300$, $r(\Sigma) = 5$.

## 5.1 Missing Values

We compared our method to popular imputations methods: `KNNImputer` (KNNI), which imputes the missing values based on the k-nearest neighbours [39], and `IterativeImputer` (II), which is inspired by the R package `MICE` [42], as coded in `sklearn` [30]; and two recent GANs-based imputation methods `MIWAE` [28] and `GAIN` [47] as found in the package `hyperimpute` [15]. The deep methods were tested using the same architectures, hyperparameters and early stopping rules as their respective papers.

In Figures 2, 3 and Table 1, we compare our estimator `MV` defined in (6) to these imputation methods combined with the usual covariance estimator on synthetic data (see App. A for details of data generation) in terms of statistical accuracy and execution time. First, `MV` beats all other methods in low-dimensional scenarios and maintains a competitive edge with `II` in high-dimensional situations when the missing data rate remains below 30%. Furthermore, it stands as the second-best choice when dealing with missing data rates exceeding 35%. Next, `MV` has by far the smallest execution time down several orders of magnitude while the execution time of `II` increases very quickly with the dimension and can become impractical (see Figure 9 for a dataset too large for `II`). Overall, the procedures `MV` and `II` perform better than `MIWAE` and `GAIN` in this experiment. Our understanding is that `MIWAE` and `GAIN` use training metrics designed to minimize the entrywise error of imputation. We suspect this may be why their performances for the estimation of covariance with operator norm are not on par with other minimax methods. An interesting direction would be to investigate whether training `MIWAE` and `GAIN` with different metrics may improve the operator norm performance.

We refer to App. E for more experiments in the MAR setting of [28, Annex 3] which led to similar conclusions. These results confirm that imputation of missing values is not mandatory for accurate estimation of the covariance operator. Another viable option is to apply a debiasing correction to the empirical covariance computed on the original data containing missing values. The advantage of this approach is its low computational cost even in high-dimension.

Table 2: We consider contaminated data following model ([2]) contaminated with a Dirac contamination of high intensity with $\varepsilon = 1$ and for several values of $\delta$ in a grid. For each $\delta$, we average the proportion of real data $\hat{\delta}$ and contaminated data $\hat{\varepsilon}$ after filtering over 20 repetitions. Values are displayed in percentages ($\hat{\delta}$ must be high, $\hat{\varepsilon}$ low)). STD stands for standard deviation.

| CONTAMINATION RATE $(1 - \delta)$ | TAIL CUT | | | | DDC 99% | | | | DDC 90% | | | |
|---|---|---|---|---|---|---|---|---|---|---|---|---|
| | $\hat{\delta}$ | STD | $\hat{\varepsilon}$ | STD | $\hat{\delta}$ | STD | $\hat{\varepsilon}$ | STD | $\hat{\delta}$ | STD | $\hat{\varepsilon}$ | STD |
| 0.1 % | 99.6 | 0.023 | 0.000 | 0.000 | 99.1 | 0.029 | 0.000 | 0.000 | 94.8 | 0.054 | 0.00 | 0.00 |
| 1% | 98.8 | 0.027 | 0.000 | 0.000 | 98.2 | 0.037 | 0.000 | 0.00 | 94.3 | 0.102 | 0.00 | 0.00 |
| 5% | 94.9 | 0.013 | 0.000 | 0.000 | 94.6 | 0.018 | 0.000 | 0.000 | 91.8 | 0.060 | 0.00 | 0.000 |
| 10% | 90.0 | 0.004 | 0.000 | 0.000 | 89.9 | 0.016 | 0.00 | 0.000 | 88.2 | 0.109 | 0.000 | 0.000 |
| 20% | 80.0 | 0.000 | 20.0 | 0.000 | 80.0 | 0.003 | 0.017 | 0.035 | 79.4 | 0.035 | 0.009 | 0.022 |
| 30% | 70.0 | 0.000 | 30.0 | 0.000 | 70.0 | 0.001 | 3.48 | 2.19 | 69.9 | 0.015 | 2.930 | 2.31 |

## 5.2 Cell-wise contamination

**Methods tested.** Our baselines are the empirical covariance estimator applied without care for contamination and an oracle which knows the position of every outlier, deletes them and then computes the `MV` bias correction procedure ([6]). In view of Theorems [1] and [1], this oracle procedure is the best possible in the setting of cell-wise contamination. Hence, we have a practical framework to assess the performance of any procedure designed to handle cell-wise contamination.

The SOTA methods in the cell-wise contamination setting are the `DI` (Detection-Inputation) method [32] and the `TSGS` method (Two Step Generalised S-estimator) [2]. Both these methods were designed to work in the standard setting $n > p$ but cannot handle the high-dimensional setting as we already mentioned. Nevertheless, we included comparisons of our methods to them in the standard setting $n > p$. The code for `DI` and `TSGS` are from the R packages `cellwise` and `GSE` respectively.

We combine the `DDC` detection procedure [34] to first detect and remove outliers with several estimators developed to handle missing values. Our main estimators are `DDCMV` (short for Detecting Deviating Cells Missing Values), which uses first `DDC` and then computes the debiaised covariance estimator ([6]) on the filtered data, and `tailMV`, which detects outliers through thresholding and then uses again ([6]). But we also proposed to combine the `DDC` procedure with imputation methods `KNNI`, `II`, `GAIN` and `MIWAE` and finally compute the standard covariance estimator on the completed data. Hence we define four additional novel robust procedures which we call `DDCKNN`, `DDCII`, `DDCGAIN` and `DDCMIWAE`. To the best of our knowledge, neither the first approach combining filtering with debiasing nor the second alternative approach combining filtering with missing values imputation have never been tested to deal with cell-wise contamination. A detailed description of each method is provided in App. [C].

**Outlier detection and estimation error under cell-wise contamination on synthetic data.** We showed that the error of a covariance estimator under cell-wise contamination depends on the proportion of remaining outliers after a filtration. In Table [2] we investigate the filtering power of the `Tail Cut` and `DDC` methods in presence of Dirac contamination. We consider the cell-wise contamination setting ([2]) in the most difficult case $\varepsilon = 1$ which means that an entry is either correctly observed or replaced by an outlier (in other words, the dataset does not contain any missing value). For each values of $\delta$ in a grid, the quantities $\hat{\delta}$ and $\hat{\varepsilon}$ are the proportions of true entries and remaining contaminations after filtering averaged over 20 repetitions. The DDC based methods are particularly efficient since the proportion of Dirac contamination drops from $1 - \delta$ to virtually 0 for any $\delta \geq 0.74$. In Figures [1] and [4], we see that the performance of our method is virtually the same as the oracle `OracleMV` as long as the filtering procedure correctly eliminates the Dirac contaminations. As soon as the filtering procedure fails, the statistical accuracy brutally collapses and our `DDC` based estimators no longer do better than the usual empirical covariance. In Table [8] and Figure [5], we repeated the same experiment but with a centered Gaussian contamination. Contrarily to the Dirac contamination scenario, we see in Figure [5] that the statistical accuracy of our `DDC` based methods slowly degrades as the contamination rate increases but their performance remains significantly better than that of the usual empirical covariance.

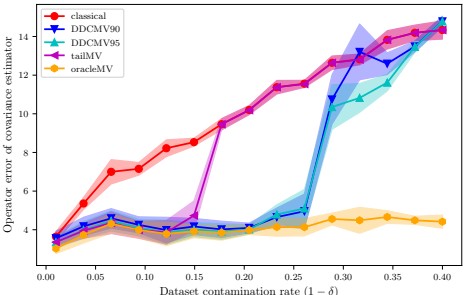

Figure 4: Estimation error as a function of the contamination rate for $n = 500$, $p = 400$, $\mathbf{r}(\Sigma) = 5$ and Dirac contamination .

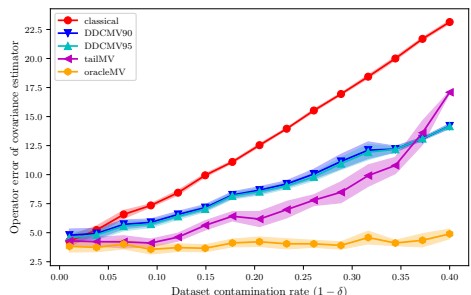

Figure 5: Estimation error as a function of the contamination rate for $n = 500$, $p = 400$, $\mathbf{r}(\Sigma) = 5$ and Gaussian contamination .

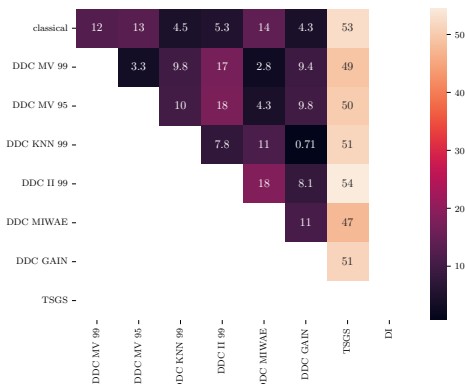

Figure 6: DI fails on ATTEND since the covariance matrix is approximately low rank. The dataset has only 8 features and the effective rank of its covariance matrix is below 2.

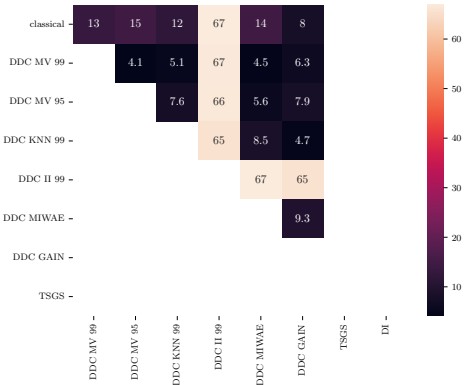

Figure 7: Woolridge's CEOSAL dataset fails both TSGS and DI with its dimension of 13 and effective rank of around 2.5.

## 5.3 The effect of cell-wise contamination on real-life datasets

We tested the methods on 8 datasets from sklearn and Woolridge's book on econometrics [46]. These are low dimensional datasets (less than 20 features) representing various medical, social and economic phenomena. We also included 2 high-dimensional datasets. See App. B for the list of the datasets.

One interesting observation is that the instability of Mahalanobis distance-based algorithms is not limited to high-dimensional datasets. Even datasets with a relatively small number of features can exhibit instability. This can be seen in the performance of DI on the Attend dataset, as depicted in Figure 6, where it fails to provide accurate results. Similarly, both TSGS and DI fail to perform well on the CEOSAL2 dataset, as shown in Figure 7, despite both datasets having fewer than 15 features.

On the Abalone dataset, once we have removed 4 obvious outliers (which are detected by both DDC and the tail procedure), all estimators reached a consensus with the non-robust classical estimator, meaning that this dataset provides a ground truth against which we can evaluate and compare the performance of robust procedures in our study. To this end, we artificially contaminate 5% of the cells at random in the dataset with a Dirac contamination and compare the spectral error of the different robust estimators. As expected, TSGS and all our new procedures succeed at correcting the error, however DI becomes unstable (see Table 3). DDC MIWAE is close to SOTA TSGS for cellwise contamination and DDC II performs better. We also performed experiments on two high-dimensional datasets, where our methods return stable estimates of the covariance (DDCMV99 and DDCMV95 are within $\approx 3\%$ of each other) and farther away from the classical estimator (See Figures 8 and 9 ). Note also that DDCII's computation time explodes and even returns out-of-memory errors due to the high computation cost of II that we already highlighted in Table 1.

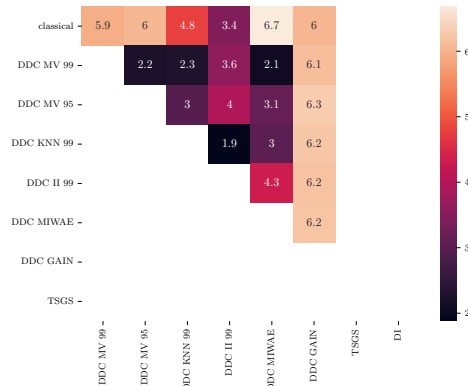

Figure 8: Relative spectral difference (in %) between covariance estimators on SP500 stock returns over 2021 and 2022. On high-dimensional data, `DDCII` becomes inconsistent with the other procedures, maybe because `II` does not scale well with dimension.

Figure 9: Relative spectral difference (in %) between between covariance estimators on NASDAQ stock returns over 2021 and 2022. Here, `DDCII` fails due to out-of-memory errors.

Table 3: Relative spectral difference (in %) between estimated covariance matrices on Abalone with 5% synthetic contamination ($\delta = 0.95$, $\varepsilon = 1$). On the cleaned dataset, all the robust estimators are very close to the empirical covariance (relative differences $< 5\%$), so we consider the empirical covariance matrix as the truth. Here the `DI` procedure fails probably due to numerical errors.

| relative error to | Classical estimator | DDCMV99 | DDCMV95 | DDC II | DDC KNN | DDC MIWAE | DDC GAIN | TSGS | DI |
|---|---|---|---|---|---|---|---|---|---|
| Truth | 12.8 | 4.12 | 6.81 | **1.70** | 2.06 | 3.46 | 5.06 | 3.06 | 8.85 |
| *std* | *0.45* | *0.29* | *0.26* | ***0.10*** | *0.092* | *0.18* | *0.35* | *0.21* | *1.48* |
| Classical | - | 13.1 | 14.3 | 13.0 | 13.0 | 12.9 | 13.1 | 13.4 | 14.9 |
| DDCMV99 | - | - | 2.99 | 2.52 | 2.22 | 1.87 | 2.66 | 5.44 | 8.79 |
| DDCMV95 | - | - | - | 5.27 | 5.03 | 4.04 | 3.71 | 8.28 | 9.99 |
| DDC II | - | - | - | - | 0.465 | 1.88 | 3.49 | 3.27 | 8.28 |
| DDC KNN | - | - | - | - | - | 1.58 | 3.19 | 3.46 | 8.15 |
| DDC MIWAE | - | - | - | - | - | - | 1.70 | 4.50 | 7.22 |
| DDC GAIN | - | - | - | - | - | - | - | 5.97 | 6.71 |
| TSGS | - | - | - | - | - | - | - | - | 6.94 |

## 6 Conclusion and future work

In this paper, we have extended theoretical guarantees on the spectral error of our covariance estimators robust to missing data to the missing at random setting. We have also derived the first theoretical guarantees in the cell-wise contamination setting. We highlighted in our numerical experimental study that in the missing value setting, our debiased estimator designed to tackle missing values without imputation offers statistical accuracy similar to the SOTA `IterativeImputer` for a dramatic computational gain. We also found that SOTA algorithms in the cell-wise contamination setting often fail in the standard setting $p < n$ for dataset with fast decreasing eigenvalues (resulting in approximately low rank covariance), a setting which is commonly encountered in many real life applications. This is due to the fact that these methods use matrix inversion which is unstable to small eigenvalues in the covariance structure and can even fail to return any estimate. In contrast, we showed that our strategy combining filtering with estimation procedures designed to tackle missing values produce far more stable and reliable results. In future work, we plan to improve our theoretical upper and lower bounds in the cell-wise contamination setting to fully clarify the impact of this type of contamination in covariance estimation.

**Acknowledgements.** This paper is based upon work partially supported by the Chaire *Business Analytic for Future Banking* and EU Project ELIAS under grant agreement No. 101120237.

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
