# OpenReview forum: "Robust covariance estimation with missing values and cell-wise contamination"
_NeurIPS.cc/2023/Conference — NeurIPS 2023 poster_

### Official Review · Reviewer_PVnK · 2023-06-24

**Soundness:** 3 good
**Presentation:** 2 fair
**Contribution:** 3 good
**Rating:** 6
**Confidence:** 4

**Summary:**

The paper “Robust covariance estimation with missing values and cell-wise contamination”  presents an unbiased estimator for covariance in the presence of missing values that do not require any imputation step, and the authors claim they achieve competitive results compared to other state-of-the-art methods. The authors check their findings in synthetical and real-world datasets and compare them to other methods such as imputation using KNN or the MICE algorithm

**Strengths:**

The notion of rigorously establishing and deriving bounds for covariance matrix estimation in the presence of missing data is intriguing, as it suggests that the bounds established for covariance estimation in complete data scenarios should extend to account for missing data. Hence, this contribution from the paper opens up new avenues of thought for researchers in the field of "missing data.”

**Weaknesses:**

However, there are some substantial flaws in the paper that I would like to point out. First and probably most important, I consider the paper is not well structured and it is challenging to follow, even for experienced readers in the field of missing data handling. There is no evident thread from which the reader can follow the ideas presented throughout the paper. The connection between sections, paragraphs and ideas is often misleading: the reader might be introduced to new notations, and new concepts, or expected to remember previous concepts that were not actually introduced (Line 108 you comment $\Sigma$ is low rank with no extra justification). Specially, after reading the paper the ultimate goal of the paper still remains unclear: is it providing a novel method for covariance estimation under missing data, imputing missing data directly?

The paper also lacks a clear section presenting the related work. These sections are often included in papers because: 1) they permit newcomers to the field to get some background, 2) they set the presented paper in the context of other historical or concurrent works, and outline the clear differences w.r.t others and 3) they allow experienced readers to corroborate if the proposed paper has properly undergone a prior deep literature review. I feel this paper lacks this important aspect: authors claim they compare to SOTA methods, but the most recent paper dates from 2021, then 2020, and after that 2018. Although MICE and KNN imputation are widely used in the missing data literature, the missing data is a rather hot topic in the machine learning field, and many works have proposed novel methods in the context of VAEs([1], [2], [3], [4]), GANs([5]) or recently diffusion models ([6]). I consider claiming this paper can outperform SOTA methods might be a strong assertion.

This, and the lack of comparison with standard datasets used in the missing data literature (tabular datasets from UCI or high-dimensional datasets such as MNIST, and CelebA, for example) makes the experimental section difficult to follow. The synthetic and real-world datasets are not described at all in the main manuscript and are referred to the Appendix, but the authors make comments about these datasets in the main text. From Figures 4 and 5 it is still not evident to me which method they propose, because it seems like they mix already proposed methods with their new covariance estimation. What’s more, the authors refer to the Figures and Tables in the Appendix and make comments about the results in the main text. I believe if comments about Figures are to be made, then these figures should be added in the main manuscript; or conversely, comments should be made in the Appendix. Something in between might make the reading a bit cumbersome.

[1] Nazabal, A., Olmos, P. M., Ghahramani, Z., & Valera, I. (2020). Handling incomplete heterogeneous data using vaes. *Pattern Recognition*, *107*, 107501.

[2] Mattei, P. & Frellsen, J.. (2019). MIWAE: Deep Generative Modelling and Imputation of Incomplete Data Sets. Proceedings of the 36th International Conference on Machine Learning, in Proceedings of Machine Learning Research 97:4413-4423 Available from https://proceedings.mlr.press/v97/mattei19a.html.

[3] Ma, C., Tschiatschek, S., Palla, K., Hernández-Lobato, J. M., Nowozin, S., & Zhang, C. (2018). Eddi: Efficient dynamic discovery of high-value information with partial vae. *arXiv preprint arXiv:1809.11142*.Chicago

[4] Ma, C., Tschiatschek, S., Turner, R., Hernández-Lobato, J. M., & Zhang, C. (2020). VAEM: a deep generative model for heterogeneous mixed type data. *Advances in Neural Information Processing Systems*, *33*, 11237-11247.

[5] Yoon, J., Jordon, J., & Schaar, M. (2018, July). Gain: Missing data imputation using generative adversarial nets. In *International conference on machine learning* (pp. 5689-5698). PMLR.

[6] Zheng, S., & Charoenphakdee, N. (2022). Diffusion models for missing value imputation in tabular data. Representation Learning Workshop at NeurIPS.

**Questions:**

I still have some questions that could maybe help to make the paper more easy to understand:

1. In Equation 2, why $X_i, \psi_i, d_{i,j}, e_{i,j}$ are mutually independent? Isn’t this a rather strong assumption?
2. The oracle using the MV method seems to always perform perfectly. More comments on this would be helpful to understand why. My first believe is that there might not be a fair comparison w.r.t the other methods.
3. In Table 2: What are DDCMV90, DDCM95? I cannot find where these are defined.
4. What is the so called “classical” covariance estimator? There is no definition of such method.

**Limitations:**

The authors don’t make any comment about the limitations of their method. However, I found a rather strong limitation in Appendix B. The method is tested on real-world datasets from sklearn, but the authors remove the categorical variables and variables that might be multimodal. This is a strong limitation since the method is tested on very smooth data distributions. I believe the authors should perform experiments in these datasets, assuming the natural artefacts that might appear.

The lack of a clear thread that makes the narrative of the paper easy to follow, the difficulty of understanding the final results from the experimental section because some figures do not appear and the need for further comparison of the proposed method with more SOTA models and other datasets drive me to believe that this paper lies below the acceptance threshold for NeurIPS conference at this point. I also hope some of my concerns can be addressed during the rebuttal period.

---

> ### Author Rebuttal · Authors · 2023-08-09
>
> We thank the referee for the detailed feedback.
>
> We would like to address first the weaknesses pointed out by the referee.
>
> - The approximately low-rank assumption is standard in the literature on the theoretical analysis of covariance estimation. For many datasets including numerous UCI datasets, the covariances are approximately low-rank.
>
> -  _Purpose of our paper_. We certainly agree that robust covariance estimation with missing data is a different problem from data imputation. We will complete our literature discussion with the references the referee kindly provided and a message along these lines:
>
> “Data imputation is a fundamental field of research with the development of sophisticated solutions based on deep learning, GANs, VAE or Diffusion schemes [1,2,3,4,5] to perform complex tasks like artificial data generation or image inpainting. While the need for efficient imputation methods is undeniable for these applications, for the specific task of covariance estimation, we propose a simple, user-friendly debiasing scheme requiring no imputation and attaining the minimax optimal rate.”
>
> We will be happy to complete this discussion as the referee sees fit regarding the data imputation literature.
>
> We confirm that our goal is to study the robust covariance estimation problem when the dataset contains missing values and/or cell-wise contaminations. Hence we discussed SOTA methods for this specific problem. We do not treat in our work the fundamental problem of data imputation and we honestly believe that our paper entitled “Robust covariance estimation with missing values and cell-wise contamination” was not misleading on our purpose and contributions which we enumerated in the global response to the referees.
>
>
>
>
> - _Lack of comparison with standard datasets used in the missing data literature_. In Appendix B Table 5, we describe the 10 real datasets of various sizes and taken from diverse domains that we used in our benchmark. Note that 3 datasets taken from sklearn are actually from the UCI database. We could not place this description in the main body. Indeed, given the 9 pages limitation and our main objective to present our theoretical findings, we made the choice to place this description in Appendix B as it is quite common for theoretical papers submitted at ML conferences.
>
> - _Compared methods in Figures 4 and 5_. Our practical method for cell-wise contamination comprises 2 steps: 1st. filtering the dataset 2nd applying Estimator in Eq (5) called MV as we described on line 212. We applied different filtering procedures in combination with MV in Fig 4 and 5 as we described in lines 240-243.
>
> - _Discussion of additional material in Appendix_. We agree. In the revision, we will move the comments next to their figures in Appendix whenever required.
>
>
> We address now the questions raised by the referee.
>
> - Q1. These assumptions are standard in the literature on cell-wise contamination like in [Agostinelli et al. (2014), Eq. (2)], especially when we investigate the theoretical properties of this problem. Note that these assumptions are relaxed in Theorem 4 where the contaminations $\xi$ admit arbitrary distributions.
>
> Agostinelli, Leung, Yohai, and Zamar (2014). Robust estimation of multivariate location and scatter in the presence of cellwise and casewise contamination
>
> - Q2. This may not be easy to see with the naked eye, but the performance of oracleMV does degrade as the contamination parameter increases but slower than for the other methods. Hence it may appear constant for contamination probability in the range $(0, 0.4)$. In the rebuttal pdf file, we included a figure where we extended the x-axis with contamination probability ranging from $0$ to $1$. We can see more clearly that the performance of oracleMV deteriorates as the contamination probability increases as we should expect from theory.
> Note however that in the literature on robust estimation with contamination, it is proved that this problem is hopeless when the contamination probability exceeds 50% since in that case, the true observations are in minority. This explains why we did not go beyond the range (0, 0.5) in our Exps. In practice, methods capable of handling 20% or more cellwise contaminations are considered quite good and of practical interest. Remarkably, our DDCMV method is doing significantly better than the classical sample covariance estimator for contamination probabilities up to 35%. Conversely, other methods experience breakdowns at contamination levels between 15% and 18%.
>
> - Q3. As described on lines 240-243, we implemented the DDC filtering combined with MV (Eq 5). 90% or 95% correspond to the trimming parameter of the DDC filtering method.
>
>
> - Q4. This is the classical empirical covariance estimator that we introduced on line 117:
> $\hat{\Sigma}^Y = \sum_{i=1}^n Y_i \otimes Y_i/n$
>
>
> Regarding the limitation on quantitative features raised by the referee, we would like to clarify that our paper concerns robust covariance estimation, primarily with applications like PCA in mind. Therefore, we focused solely on quantitative features. For studying relationships between quantitative and categorical features (e.g., color of the eyes), alternative methods like ANOVA tests would be more appropriate, as computing the covariance between a quantitative and a categorical covariate does not make sense.

---

> > ### Comment · Reviewer_PVnK · 2023-08-15
> >
> > First of all I would like to thank the authors for the detailed answers, not only to me, but also to the other reviewers. In the following I answer the rebuttal from the authors:
> >
> > ### Goal of the paper
> > I consider that the goal of the paper should be stated clearer, and I hope the authors can do it properly in the next phases if necessary. I agree that a clear distinction between missing data imputation and covariance estimation should be made at the very beginning of the paper. Especially, because the research in missing data imputation is gaining more and more importance, and this paper could be read by practitioners in this area of research. Bridging both areas is rather important for this paper, I believe.
> >
> > ### Comparison to other datasets
> > I cannot find any Table 5 related to UCI datasets results. I see Table 4 describes the UCI datasets, and Appendix H and Figures from the last pages provide some results. I think some of these results should be added to the main paper. Or proper references should be made. I mean, in Section 5.3., __in the main paper__ you comment about results from Figures in the Appendix. I do not understand this section. I think it is easier for the reader if you incorporated the main results from the real datasets in the main manuscript and comment it there, or just leave everything in the Appendix. But adding Figure references in the main paper that refer to Figures in Appendix is very cumbersome.
> > What's more, I already mentioned this in the first review, and from the rebuttal it seems that the authors did not understand my point. Please let me know if you need more clarification.
> >
> > ### Missing data imputation strategies
> > Authors claim both in the paper and the rebuttal that they address a different problem from missing data imputation, namely __covariance estimation__. However, they compare to two __missing data imputation__ methods implemented in sklearn: knn imputer, and iterative imputer, based on the MICE algorithm. This is counterintuitive to me regarding the author's rebuttal. I do not understand why the authors do not find it necessary to compare SOTA imputation methods using DGMs for example, or any SOTA methods presented in recent years at ML conferences, and decide to compare to those baseline methods.
> >
> > This point still concerns me, because although the aim of the papers is covariance estimation, I believe there is much overlap with missing data imputation, and more modern methods could have been tested. This does not minimize the strong theoretical derivations and proofs of the paper, but I believe the experimental section should be improved to make the paper sound enough for the conference. Besides, the IterativeImputer (MICE [1]) is al algorithm from 2011. Although very powerful, I belive the authors should compare ito more recent methods.
> >
> > ### Figure for oracleMV
> > I thank the authors for the new figure included in the rebuttal, and I think it could be included in the Appendix and properly referenced in the main manuscript. It is very interesting and I guess it can answer a common question for any reader of this paper.
> >
> > ### Categorical Variables
> > I agree that the paper is not primarily focused on dealing with mixed-type data. However, the authors should comment on the filtering out of those categorical variables, since tabular real-world datasets such as the UCI datasets are corrupted by artefacts and are composed of variables of different statistical types. Other researchers who have worked with these datasets would also like to know how the proposed method was actually tested on these datasets. A detailed description of the setup should be included in the main manuscript or properly referenced at the beginning of Section 5.3.
> >
> > Finally, I appreciate the hard work of the authors. However, I believe the comparison to SOTA methods based on DGMs should be included since those methods are actually the SOTA methods used __at least__ in the missing data imputation community, not only KNN Imputer and MICE, and comparing to most SOTA methods is a must-have property for a paper at this conference. I maintain my score and I await further answers from the authors and other reviewers, and still consider the possibility to increase the score during the discussion period.
> >
> >
> > # References
> > [1] Van Buuren, S., & Groothuis-Oudshoorn, K. (2011). mice: Multivariate imputation by chained equations in R. Journal of statistical software, 45, 1-67.

---

> > > ### Author Response · Authors · 2023-08-16
> > >
> > > We are greatly grateful to the referee for the detailed feedback on our work.
> > >
> > > We apologize for the extended length of our response, leading us to divide it into three parts (including experiments tables). However, it was necessary to thoroughly address all the raised points by the Referee.
> > >
> > > __Goal of the paper.__
> > > We agree with the referee’s recommendation. To provide a more comprehensive context for our contributions, we propose to do the following in the revision.
> > >
> > > First, we will complete the second introduction paragraph (lines 32-43) with relevant references on the more advanced imputations techniques including those provided by the referee.
> > >
> > > Next, we will add in the introduction a paragraph that bridges our robust matrix estimation work with imputation literature, presenting the following key points:
> > > (i) Understanding the theoretical aspects of learning covariance under missing values and cellwise contamination holds significant significance. Our newly derived minimax learning rates offer insights into these dimensions.
> > > (ii) We will introduce our two strategies, distinct from existing methods in covariance estimation with cellwise contamination. The first approach involves __filtering+ debiasing__ (no imputation), while the second entails __filtering+ imputation__ followed by standard covariance estimation. As suggested by the referee, we will explicitly clarify our focus on covariance estimation concerning missing values and cellwise contamination. Our objective is to compare these two strategies, emphasizing the trade-off between statistical accuracy (in terms of operator norm deviation) and computation time, as well as potential ease of use.
> > >
> > > Our current benchmark contains the debiasing scheme and the filtering + baseline imputations (KNN, Iterative Imputer) that already beat the SOTA in covariance estimation with cellwise contaminations: TSGS and DI (References [1] and [21] in our paper). But we agree with the referee’s that including other SOTA imputation methods is necessary to strengthen our study. We are currently adding __GAIN__ [Yoon et al. ICML 2018] and __MIWAE__ [Mattei and Frellsen, ICML 2019] to our benchmark. We use the implementation provided by the HyperImpute library and the code provided by the authors of the __MIWAE__ method on their github.
> > >
> > > Based on the current theoretical results and novel benchmark results, our discussion will go along this way:
> > > - For the MCAR values problem on synthetic data, __debiasing__ is similar to IterativeImputer in terms of operator norm statistical accuracy but much faster to compute and extremely simple to use in our synthetic benchmark. We have also implemented __MIWAE__ and __GAIN__ both with the same architectures and choices of hyperparameters for the UCI database__ as they proved to work well in their respective papers. The statistical performances of __MIWAE__ are close to that of KNNimputer for a computation time 20 times longer than IterativeImputer. __GAIN__ accuracy is close to that of IterativeImputer and MV for missing value probability smaller than 0.3 and for a computation time 3 times longer than IterativeImputer. For missing values probability larger than 0.3, __GAIN__ performance is closer to that of KNN imputer.
> > > - Our understanding is that __MIWAE__ and __GAIN__ use training metrics designed to minimize the entrywise error of imputation. We suspect this may be why their performances for the covariance estimation with operator norm are not on par with other minimax methods. This is a compelling question to study whether training __MIWAE__ and __GAIN__ with different metrics may improve the operator norm performance.
> > > - To complete our comparison on missing values, we replicated the experiment with the MAR mechanism (but not MCAR) of [Mattei and Frellsen, ICML 2019, supplementary Section 3] where it is expected that methods __MIWAE__ and __GAIN__ perform better than baseline competitors. We also applied our debiasing scheme with heterogeneous missingness estimation (Please see our reply to Reviewer zw68 for the formula that will be integrated in the revision).  On the Abalone dataset, IterativeImputer is the most accurate for the operator norm. __MIWAE__ and __GAIN__ are far behind the second best __MV__. The Breast Cancer data was used both in the MIWAE and GAIN papers. We compared __MIWAE__ and __GAIN__ to our methods on this dataset. We used the colab code provided by the authors of __MIWAE__ to rerun their experiment. For __GAIN__ we use the defaults parameters as in the HyperImpute library. __GAIN__ is the second best method behind MV and is better than IterativeImputer. On the other hand, in all our experiments, the computation time is far longer for MIWAE and GAIN  than for our debiasing scheme MV.

---

> > > > ### Author Response · Authors · 2023-08-16
> > > > **continued answer**
> > > >
> > > > Table 1: Abalone with the MAR setting of [Mattei and Frellsen, 2029]. We represent the relative spectral error in percentages.
> > > > |       | classical | MV | II | KNN | MIWAE |  GAIN |
> > > > |------:|----------:|----------:|----------:|----------:|-----------:|---------|
> > > > | Truth | 59.230366 |  3.380323 |  1.928303 |   2.237872 |  8.407057 | 16.594489 |
> > > >
> > > >
> > > > Table 2: Same with Breast Cancer
> > > > |       | classical | MV  | II  | KNN  | MIWAE |  GAIN |
> > > > |------:|----------:|----------:|----------:|----------:|-----------:|---------|
> > > > | Truth | 88.492862 | 31.388347 | 41.040478 |  43.262434 | 88.123771 | 40.168051 |
> > > >
> > > > Execution time on a synthetic dataset of size p=50, n=300 (in seconds)
> > > > |       | mean time | std      |
> > > > |-------|-----------|----------|
> > > > | MV    |  0.000308 | 3.60e-05 |
> > > > | KNN   |   0.0239  |  0.00873 |
> > > > | II    |    2.33   |   1.04   |
> > > > | GAIN  |    6.94   |   0.480  |
> > > > | MIWAE |    50.5   |   2.80   |
> > > >
> > > > - In the cellwise contamination setting of Table 3 (in our paper) on Abalone dataset, we have added methods __filtering +MIWAE__ and __filtering + GAIN__  Below is the first line containing the most important information. certain sections of the code within __MIWAE__ and __GAIN__ utilize a random generator, which can lead to slight performance variations compared to the results presented in Table 3 of the paper (even when run on the same seed). However the order of magnitudes are preserved and our conclusions remain consistent. Here __DDC MIWAE__ is on par with SOTA TSGS for cellwise contamination and __DDC II__ does better.
> > > >
> > > >
> > > > |       |  Classical  | DDC MV 99 | DDC MV 95 | DDC II 99 | DDC KNN 99 | DDC MIWAE | DDC GAIN |     TSGS |        DI |
> > > > |------:|----------:|----------:|----------:|-----------:|----------:|---------:|---------:|---------:|-----------|
> > > > | Truth | 13.097475 |  4.205185 |  6.548251 |   1.928303 |  2.237872 |  3.37452 | 5.468142 | 3.230577 | 35.137091 |
> > > >
> > > >
> > > > In the revision, we will complete Figures 1 through 6 and Table 3 with the methods __MIWAE__ and __GAIN__ and __DDC +MIWAE__ and __DDC + GAIN__ .
> > > >
> > > > - In the cellwise contamination setting, we also ran __DDC+MIWAE__ and __DDC + GAIN__ on the high-dimensional dataset SP500 that TSGS and DI cannot handle. It is worth noting that DDC-MIWAE is very close to other Methods DDC MV, DDC KKN and DDC II. But DDC GAIN looks further away raising doubt on its accuracy in this experiment.
> > > >
> > > > Table 3 : Relative norm between estimated covariance matrices of the SP500
> > > >
> > > >
> > > > |           | DDC 99 | DDC MV 95 | DDC KNN | DDC II | DDC MIWAE | DDC GAIN |
> > > > |-----------|--------|-----------|---------|--------|-----------|----------|
> > > > | classical |   5.9  |    6.0    |   4.8   |   3.4  |    6.7    |    6.0   |
> > > > | DDC MV 99 |        |    2.2    |   2.3   |   3.6  |    2.1    |    6.1   |
> > > > | DDC MV 95 |        |           |   3.0   |   4.0  |    3.1    |    6.3   |
> > > > | DDC KNN   |        |           |         |   1.9  |    3.0    |    6.2   |
> > > > | DDC II    |        |           |         |        |    4.3    |    6.2   |
> > > > | DDC MIWAE |        |           |         |        |           |    6.2   |
> > > >
> > > > Based on these additional experiments, we see the debiasing scheme as a reliable baseline method that could provide accurate preliminary information (if not the best) very fast even in the MAR and cellwise contaminiatin settings. We can of course run a more advanced imputation method like GAIN or MIWAE for a potential gain in operator norm accuracy but for a much longer computation time.
> > > >
> > > > Our current opinion is that the difficulty of the problem should naturally depend on the data structure and the missingness/contamination assumptions. As the problem becomes harder and depending on the learning task and desired accuracy, it may become necessary to use more complex imputation methods at the cost of significantly longer computation time.
> > > >
> > > > In the conclusion section, we will discuss as a perspective for future work that (1) combining robust statistical techniques and advanced imputation methods is a promising line of research to handle missing values and contaminations, (2) developing the minimax framework for other learning tasks will provide a useful theoretical benchmark to assess the difficulty of a learning task with missing values and contaminations and thus advance the theoretical understanding of these problems for other learning tasks.

---

> > > > > ### Author Response · Authors · 2023-08-16
> > > > > **Continued answer (last part)**
> > > > >
> > > > > __Comparison to other datasets.__
> > > > >
> > > > > We meant Table 4 and not Table 5 for the list of datasets included in our experiments. We are sorry for the confusion. We understand that the 3 datasets Abalone, Breast cancer and Wine are originally from the UCI database and have been included in the list of datasets available on sklearn. It may actually be more fair to specify they are originally from UCI database in Table 4. We will do it in the revision.
> > > > >
> > > > > __Devoting more space to the experimental section 5 in the main paper.__
> > > > >
> > > > > Since the paper is of theoretical nature, we would like to keep theoretical bounds in the main paper. But we shall move the proof arguments to the appendix to get more space for the experimental section. This should allow us to display Figures 6 and 11 as well as Table 5 in the main paper. We will also include in our current Figures and Tables the __GAIN_ and _MIWAE_ imputation methods.
> > > > >
> > > > > __Missing data imputation strategies.__
> > > > >
> > > > > We agree with the referee. Please see our response above to goal of paper.
> > > > >
> > > > > __Figure for oracleMV.__
> > > > >
> > > > > Thanks for the suggestion! Actually, we were also planning to add this figure to the Appendix for the discussion of the missing values problem.
> > > > >
> > > > > __Categorical Variables.__
> > > > >
> > > > > To improve clarity, we propose to add one sentence discussion at the beginning of Section 5.3 to justify our choice of filtering out categorical variables as we focus on the covariance estimation. We will also point out to Appendix B for more details on the preprocessing.
> > > > >
> > > > >
> > > > >
> > > > >
> > > > > Finally, we express our gratitude to the Referee for the engaging discussion that has opened new promising perspectives at the interface between the fields of Robust statistics and Imputation for our work. We hope that the suggested revisions effectively address the pertinent inquiries raised by the referee. We remain at disposal for further discussion or to address any additional concerns the referee may have.

---

> > > > > > ### Comment · Reviewer_PVnK · 2023-08-19
> > > > > > **Great work, update score**
> > > > > >
> > > > > > I would like to thank the authors for their responses and for their hard work on this rebuttal period.
> > > > > >
> > > > > > First, I am grateful for performing additional experiments with some suggested methods. I think now the papers can be also appealing yo anyone interested on missing data handling. I believe the inclusion of these last results greatly improve the paper. As a last suggestion, try to perform different run for different seeds and get std errors for the table results.
> > > > > >
> > > > > > Also, be careful to properly include comments on why SOTA methods such as MIWAR or GAIN might be failing/being better than your method. Being transparent about this, both when the proposed method outperforms or fails, is key point, and actually demonstrates the through understanding of the problem from the authors. I agree the problem depends fully on the data structure and the missing corruption. Also including some comments about it in the main manuscript or Appendix would be nice. In general, try to add the formatting and the other suggestions of the clarity of the paper in the revision.
> > > > > >
> > > > > > Overall, I am grateful that we could arrive to a common point in covariance estimation and missing data imputation. Thus, I increase my score. Thank you again for the hard work.

---

> > > > > > > ### Author Response · Authors · 2023-08-19
> > > > > > > **Thanks**
> > > > > > >
> > > > > > > We extend our gratitude to the referee for their valuable recommendations that significantly improved our work and their reassessment of our paper.
> > > > > > >
> > > > > > > We are finalizing the addition of standard errors to our tables, as suggested. The discussion of our experimental study in the main paper will be thoroughly updated with the results and comments from our previous response to the referee.
> > > > > > >
> > > > > > > We also realised that using several imputation methods working in different ways, offers a distinct advantage in addressing cell-wise contaminations. This allows us to compare outputs from multiple methods, enhancing the confidence in our findings. Table 3 from our previous response is a prime example, highlighting the strong agreement among four out of six robust methods, with a relative precision of approximately 3%. This is another message that we will add to our discussion in the main paper in addition to all the other points we have discussed in our previous response.

---

### Official Review · Reviewer_zw68 · 2023-07-05

**Soundness:** 3 good
**Presentation:** 3 good
**Contribution:** 3 good
**Rating:** 6
**Confidence:** 3

**Summary:**

The objective of this paper is to mitigate the influence of outliers and missing values in covariance estimation without resorting to imputations.  A particular type of missing data that satisfies assumption (1) is considered and an unbiased estimator of the true covariance  that achieves the minimax bound with the operator norm is proposed.  For the outliers, they are further assumed to be cell-wise contaminated satisfying assumption (2) and two further assumptions (see lines 169 and  173).  Under these assumptions, an unbiased estimator of the true covariance is proposed, and its upper and lower bounds in the operator norm are established.  Numerical experiments demonstrate the advantages of the proposed approaches over existing approaches in a high-dimension and low rank setting.

**Strengths:**

This is a paper with both new methodology and theory for an interesting problem. The primary contribution is its strong theoretical results, which significantly  improve over existing results.

**Weaknesses:**

The assumptions on missing values seem strong.  Usually, measurements for a subject are collected at the same time.  So, forcing the missing mechanism to be independent across the components seems to limit its applications. Also, why would all the components have the same missing probability? Some components might be harder to measure, hence more likely to be missing.  This assumption is much stronger than MCAR.

The definition of cell-wise contamination in (2) is restrictive as an outlier for a vector data does not have to be an outlier in any of the component. Also, it is further assumed in line 169 that components of cell-wise contaminations are uncorrelated.



**Questions:**

Does equation (3) only involve missing values with no cell-wise contaminations as it does not involve any of the parameters in (2)?

The \xi_k in line 86 are random vectors with diagonal covariance matrix (see line 169).    Is this diagonal assumption a key to the results in Section 4? What would happen if they were not diagonal?



**Limitations:**

The proposed approach involves several assumptions that will limit its applicability in practice.

---

> ### Author Rebuttal · Authors · 2023-08-09
>
> We thank the referee for the careful reading of our paper.
>
> We discuss first the weaknesses identified by the referee.
>
> - W1. Regarding the first identified weakness, we decided to present first the classical MCAR setting for the sake of clarity. But we also covered the heterogeneous missingness case at the end of Section 3 (line 161). In that setting, each component has a different missingness probability and we provided a statistical guarantee in Eq. 8 (See also the proof in Appendix E.4). We did not give more details due to lack of space but in that case, the debiased covariance scheme is
> $$
> \widetilde{\Sigma} = \mathrm{diag}(\delta_1^{-1},\cdots,\delta_p^{-1}) \odot \mathrm{diag}(\Sigma^Y) + A\odot  \left( \Sigma^Y - \mathrm{diag}(\Sigma^Y) \right),
> $$
> where the $p\times p$ matrix $A$ admits entries  $A_{j,j}=0$ for any $j\in [p]$ and off-diagonal entries
> $$
> A_{j,k} = \frac{1}{\delta_j\delta_k},\quad \text{if $j\neq k$}.
> $$
> Here $\odot$ stands for the (Hadamard) entrywise product.
>
> We will add more details on the heterogeneous missingness case in the appendix in the revision of the paper.
>
>
> - W2. Regarding the second identified weakness, we understand that the setting described by the referee may correspond to the classical Huber’s contamination setting that has been the focus of many studies in the literature. Our understanding is the cell-wise contamination setting and the Huber contamination settings are very different (one cannot be considered as a subcase of the other). As a consequence, methods developed for Huber contamination may not be adapted to cell-wise contamination and vice versa. We discussed this aspect on lines 25-31, but we would be happy to further precise this aspect in the revision if requested by the referee.
>
> We answer now the questions raised by the referee.
>
> - Q1. Yes we consider the estimator in Eq (5) which corresponds to the case of missing values and no contamination. Later in the paper, we consider two strategies to handle the cell-wise contamination setting. The first is the estimator in Eq. (10) that we studied from a theoretical perspective in order to determine the minimax rate of estimation in the cell-wise contamination setting. The second strategy is a combination of filtering to detect and replace all contaminated entries by a $0$ and then apply the missing values estimator Eq. (5) on the filtered dataset. This second strategy is user-friendly and produces good results in our experimental study. It beats the SOTA method for covariance estimation with cell-wise contaminations ([1,4,21,22,23]).
>
>
> - Q2. Thanks for this great question. We considered in Eq (2) the setting where the cell-wise contamination happened at random independently of each other as it is common in this literature considered a realistic assumption to model sensor failures ([1,4,21,22,23]). The consequence is that the matrix $ \Lambda $ is diagonal.
> Coming back to your question, we believe that it may be possible to cover the case of non diagonal $\Lambda$ by introducing an appropriate modification of Eq (10) and a similar (albeit more technical) analysis as the one we use to prove Theorem 3. We may not pursue this direction further as the estimator in Eq (10) was introduced solely for theory purposes to understand the minimax rate in the cell-wise contamination setting. As we mentioned above, our practical solution is rather to filter the dataset first and then apply the Missing Values estimator in Eq. (5).
>
> We will add discussion on these aspects in the revision of our paper. Thanks!

---

> ### Comment · Reviewer_zw68 · 2023-08-16
>
> Thank you very much for your response.  I appreciate the clarification but will keep my original score.

---

### Official Review · Reviewer_LnKP · 2023-07-06

**Soundness:** 3 good
**Presentation:** 3 good
**Contribution:** 3 good
**Rating:** 7
**Confidence:** 3

**Summary:**

This work proposes an unbiased estimator for the covariance matrix under missing entries. Unlike most existing approaches, the proposed estimator does not require an imputation step to discard missing values before computing the covariance matrix. Interestingly, the proposed method employs an approach that avoids matrix inversion in high dimensions, thereby overcoming large complexity issues.
Moreover, along with relevant outlier detection methods, the proposed estimator can be used used under the presence of cell-wise outliers to tackle such outliers in a high dimensional setting with inherent low rank structure. The effectiveness of the proposed method is demonstrated via experimental studies.

**Strengths:**

Derivation of non-asymptotic estimation bounds of the covariance matrix with the operator norm and matching minimax lower bounds, enabling the quantification of impact of rates of missing values rate outliers.

Traditional methods rely on matrix inversions resulting in failure in the high-dimensional setting, while the proposed avoids matrix multiplication and therefore, performs well.

Experiments demonstrate that the proposed method is more robust to cell-wise contamination than comparing state-of-the-art methods, while producing reliable estimates of the covariance in low computation duration.



**Weaknesses:**

Missing definitions of metrics -- Please see below.

**Questions:**

What is STD in Table 2?

Please provide an expression for the relative spectral difference used in Table 3.

---

> ### Author Rebuttal · Authors · 2023-08-09
>
> We thank the referee for the careful reading of our paper. As recommended by the referee, we will precise the definitions of the metrics in the revision of this paper.
>
> - Q1. STD stands for standard deviation. We will precise it in the table caption.
>
> - Q2. The formula for the relative spectral difference of matrices $A$ and $B$ is :
> $$d(A,B) = \frac{\Vert A - B \Vert}{max(\Vert A \Vert, \Vert B \Vert)}$$
> i.e. the ratio between the spectral difference and the largest spectral norm. We will add it to the revision of the paper.

---

### Official Review · Reviewer_vhsW · 2023-07-08

**Soundness:** 3 good
**Presentation:** 4 excellent
**Contribution:** 3 good
**Rating:** 7
**Confidence:** 3

**Summary:**

This paper studies the problem of estimating a covariance matrix in the presence of missing observations and/or outliers.
The authors first consider the missing values framework and theoretically analyse the unbiased covariance estimator introduced in [Karim Lounici. High-dimensional covariance matrix estimation with missing observations]. Then, in the framework (missing values + outliers), they also recommend its use in combination with cell-wise outlier detection methods such as DDC.
Finally, to support their theoretical results, they carried out some experiments to demonstrate the advantage of their approach over existing methods.

**Strengths:**

1\  It is a real pleasure to read this paper. Indeed, It is very well written, which makes it easy to understand

2\ The contributions are clear and well-supported. In addition, the theoretical results (if true - see next section) are theoretically sound because there are probability bounds on the difference between the estimated covariance and the true covariance, but more importantly there are minimax bounds.

3\ The experimental part is again very clear and seems to compare well to existing methods with the error bars and calculation time indicated.

**Weaknesses:**

1\ The contamination is assumed to be sub-Gaussian. This contrasts with the recent robust literature, which attempts to take into account any type of contamination.

2\ The probability of missing values d and the outlier epsilon are assumed to be known in the theoretical results (although I understand that a full analysis of the estimator when d and epsilon are estimated is certainly difficult).

3\  3. There are possible problems in the theorems which should be clarified (see Questions).

================
Additional remarks:

Figure 1 is never cited and explained.

I think the formal definition of the operator norm should be given in the notation part.

line 101: "The The"

line 171: "procedure procedure"


**Questions:**

My first question is more related to Theorem 1. However, as the proofs of the other theorems use sometimes the same techniques, they are also related to them.

1\ In the proof of Theorem 1 (Appendix E.2.3), why there is no absolute value on (1/d - 1/d^2) when this scalar moves outside of the norm? This could change the proof because then |(1/d - 1/d^2)| is no longer always less than 1/d

2\ In the section "Experiments", it is stated that the best oracle is the one that knows "the position of every outlier, deletes them and then computes the MV bias correction procedure". Isn't an oracle that knows "the position of every outlier" but instead of deleting them, performs a good imputation and then uses a standard estimator better?

3\ It seems that only the estimator of equation (5) is used in the experiments. Why not also use the estimator from equation (10)?

================

One final remark: My score reflects my concerns about the first question. Of course, I'm willing to raise the score if there's something I'm missing.

**Limitations:**

-

---

> ### Author Rebuttal · Authors · 2023-08-09
>
> Q1. We thank the referee for pointing out the issue in the proof. We actually found it after submitting the paper and found an alternative simpler argument leading to an improved bound.
>
> Indeed Ineq. (29) in our paper should be:
> $$
> \Vert\widehat{\Sigma} - \Sigma\Vert  \leq \delta^{-2} \Vert diag \left(\widehat{\Sigma}^Y - \Sigma^Y\right)\Vert+ \delta^{-2} \Vert\widehat{\Sigma}^Y - \Sigma^Y\Vert.
>  $$
> Next we use the Schur-Horn Theorem to get
> $$
> \Vert diag \left( \hat{\Sigma}^Y - \Sigma^Y\right) \Vert \leq \Vert\hat{\Sigma}^Y - \Sigma^Y\Vert.
> $$
> Then, using Ineq (23), we get
> $$
>             \Vert \widehat{\Sigma} - \Sigma\Vert \lesssim  \frac{\Vert\Sigma\Vert}{\delta}\left(\sqrt{\frac{{r}(\Sigma)}{n}} \lor \frac{{r}(\Sigma)}{n} \lor \sqrt{\frac{t}{n}} \lor \frac{t}{n}\right).
> $$
> We got rid of the $\log p$ in the bound making our Theorem 1 completely dimension-free and sharp minimax optimal.
>
> We corrected the proof. We also note that the expectation sign in the LHS of (23) was a typo that we corrected. Lemma 2 is no longer needed. We hope this clarifies this issue. Thanks again!
>
> Q2. Thanks for the interesting question. According to minimax theory, given the available information, it is impossible to estimate faster than a certain universal rate, regardless of the procedure used. Hence, if our dataset has missing values, imputation will not provide more information on the problem as we rely only on the observed data to impute the missing values. Therefore we should not expect a faster minimax rate of estimation after imputation than the universal rate provided by the minimax lower bound. The minimax lower bound in Thm 2 guarantees that no estimation method can achieve a faster rate than the universal rate in Eq (7). Hence an imputation method may attain the minimax rate but cannot do better. Additionally, we proved that our debiasing method achieves the minimax rate. This means that we can build an efficient minimax optimal procedure for this covariance estimation problem without imputation.
> Now, in the cellwise contamination setting, we observe that corrupted entries lead to a deterioration in the minimax estimation rate, as shown in Eq (14), when compared to the missing values setting. Hence the cellwise contamination problem is statistically more challenging than the missing values setting. This is why we propose a user-friendly procedure that identifies and removes the outliers before applying the MV procedure.  We hope this clarifies our choice of estimator in experiments on cellwise contaminations.
>
> Q3. We considered estimator (10) only for the theoretical exploration of cellwise contamination problem. Indeed (10) requires estimating probability $\epsilon$ and contamination covariance $\Lambda$. This is difficult as we may not easily tell apart contamination from signal. It seemed more convenient to us in real life applications to apply a good filtering strategy which eliminates most of the contaminations (thus making the term $\epsilon \|\Lambda\|$ negligible) followed by the MV estimator (5). This strategy is user-friendly and beats the SOTA method for covariance estimation with cell-wise contaminations ([1,4,21,22,23]) in our experimental study.
>
> We mostly agree with all 3 weaknesses pointed out by the referee but we would like to explain our choices in light of phenomena we observed in our experimental study.
>
> 1. We observe in practice that heavy-tail cellwise contaminations (high-magnitude entries) are very easily filtered out of the dataset and only inconspicuous contaminations may remain (those of the same magnitude as real data or smaller). This is why we only investigated in Thm 3 the impact of sub-gaussian contaminations on the performance of a simple procedure. In the lower bound of Thm 4, we make no assumption on the distribution of the cellwise contamination. These 2 thms reveal that even small contaminations impact the minimax rate.
>
> The literature on robust estimation is very interesting. Some contributions consider weaker assumptions like a $L_4-L_2$ condition on the distribution of $X$ and no contamination. But this requires more complex covariance estimation procedures like one based on uniform trimming in [Abdalla and Zhivotovskiy, 2022] which are often computationally intensive. It would be interesting to explore this direction both theoretically and in practice within the cell-wise contamination setting with a focus on the computational/statistical trade-off. But this is beyond the scope of this paper.
>
> Abdalla and Zhivotovskiy (2022). Covariance Estimation: Optimal Dimension-free Guarantees for Adversarial Corruption and Heavy Tails (arXiv:2205.08494)
>
> 2. We mostly agree. In a missing values setting, we can easily estimate the probability of missing values by counting the number of NA entries in a dataset and derive theoretical guarantees for it. But, in the contamination setting of Eq(2), a full theoretical analysis of our estimators including estimation of $\delta$ and $\epsilon$ is difficult. From a practical aspect, we have built estimation procedures for $\delta$ and $\epsilon$ which work reasonably well in experiments. See Tab 2 with $\epsilon=1$ which shows that we can accurately estimate $\delta$ in some cellwise contamination setting.
>
> 3.	Please see our response to Q1.
>
> Finally, thanks for pointing out some typos. We have corrected them. We also added a mention to Figure 1 in our discussion on line 257 along with Figure 4.

---

> > ### Comment · Reviewer_vhsW · 2023-08-14
> >
> > Thank you for the comprehensive response and the different modifications. I have increased my score accordingly.

---

> > > ### Author Response · Authors · 2023-08-15
> > >
> > > We are grateful to the referee for the time and attention they dedicated to reviewing our response and reevaluating our work

---

### Author Rebuttal · Authors · 2023-08-09

We wish to thank the reviewers for their insightful evaluation of our paper. We appreciate all the comments and remarks, which we will implement in our revision. We discuss here some points that were raised by several referees.

The goal of our paper is to study the robust covariance estimation problem when the dataset contains missing values and/or cell-wise contaminations. We clarify some important aspects of our contributions:

- __For the specific problem of robust covariance estimation with missing values__, we prove (in Thms 1 and 2) that a debiasing scheme without any imputation can provide an optimal estimation procedure with theoretical minimax guarantees. This minimax result firmly establishes the fact that there cannot exist any method (even one relying on data imputation) that achieves a rate faster than the minimax rate. In other words, imputation based methods may also achieve results on par with but cannot surpass the performance of our simple method in terms of minimax optimality. Furthermore, the imputation methods we implemented incurred a higher computational cost.

- For the cell-wise contamination setting, we prove theoretical results (Thm 3 and 4) on the impact of contaminations on estimation rates. We advocate that a simple solution that performs filtering of the dataset first and then applying a simple debiasing scheme for the missing values case performs better than the SOTA methods for the robust covariance estimation problem with cell-wise contaminations ([1,4,21,22,23]). As we observed in our experiments, these SOTA methods perform poorly and are numerically unstable even on simple datasets.

- On the practical aspect, our solutions are user-friendly and computationally fast.


- As requested by one referee, we added in the attached pdf file a completed Figure 4 with missing probability in the range (0,1). We can see that the accuracy of oracle MV does degrades as missing probability increases to 1 as predicted by theory.

As requested by one referee, we added in the attached pdf file a completed Figure 4 with missing probability in the range (0,1). We can see that the accuracy of oracle MV does degrades as missing probability increases to 1 as predicted by theory.

---

### Decision · Program_Chairs · 2023-09-21

**Decision:**

Accept (poster)

**Comment:**

This paper studies the task of covariance estimation in the presence of missing data and cell-wise contamination. It provides an unbiased estimator that can achieve minimax optimal statistical accuracy. In particular, the authors show that their provided estimator achieves a non-asymptotically optimal statistical error, provided that both the missing values and the contamination model follow a Bernoulli model. The authors then complement their theoretical results with comprehensive experiments in both low and high-dimensional regimes.

Four reviewers have reviewed the paper, and their overall assessment of the paper was positive. I agree with this assessment and believe the paper is a solid contribution with interesting results. I should also mention that the authors did a great job addressing the comments raised by the reviewers. I encourage the authors to address these comments and add the complementary simulations to the revised paper.